# Generative Adversarial Learning for Semi-supervised Retinal Layer Segmentation in OCT Images

Charlene Zhi Lin Ong[†,*], Asad Abu Bakar Ali [†], Jagath C Rajapakse[*]

*Abstract*—It is often challenging to obtain large number of labeled data for retinal layer segmentation in optical coherence tomography scans due to the need for expert ophthalmologists. On the other hand, huge quantities of unlabeled scans are often collected in medical centers. In this work, we propose a novel generative adversarial learning framework, GOctSeg, for semi-supervised retinal layer segmentation. GOctSeg consists of a U-Net generator, a discriminator, and a segmentor. The generation of synthetic images from synthetic labels is performed with the U-Net generator, which is used as input together with labeled B-scan patches into the Segmentor network for retinal layer boundary regression. We have also evaluated our methodology on two datasets, the Data Resource for Multiple Sclerosis and Healthy Controls and the Duke University Diabetic Macular Edema dataset and demonstrated that our method outperformed or is comparable to other state-of-the-art methods with limited labels for boundary regression. Furthermore, we investigated the performance of GOctSeg on images with low signal-to-noise ratio or with blurred boundaries and showed that our methodology remained robust. Through ablation studies, we demonstrated the utility of synthetic labels for generative learning to guide in semi-supervised retinal layer segmentation. We envision that this methodology can be used to significantly reduce the effort required to obtain labels when there is label scarcity in clinical settings.

*Index Terms*—Segmentation, Life and Medical Sciences, Machine learning, Computer vision

## I. INTRODUCTION

Retinal layer thickness is a critical imaging biomarker for both ophthalmic and nonophthalmic diseases. Retinal layer thickness can be measured noninvasively at micrometer-resolution by optical coherence tomography (OCT) [1]. The OCT scans are usually acquired in 3D volumes of B-scans, with each B-scan as a sequence of A-scans. Each A-scan, or single axial scans, indicates the composition of optical reflectance at a certain point through the tissue depth. The sequence of A-scans makes up a cross-sectional image of the retina, or B-scan, by imaging across the tissue [2]. Retinal layer thickness has been shown to be a useful diagnostic biomarker for early age-related macular degeneration, where studies have shown a statistically significant thickening of retinal pigment epithelium and a reduction of photoreceptor

[†]Charlene Zhi Lin Ong and Asad Abu Bakar Ali are with MSD International GmbH, Singapore, Singapore. Emails: {charlene.zhi.lin.ong, asad.abu.bakar.ali}@.msd.com

[*]Charlene Zhi Lin Ong and Jagath C Rajapakse are with Health Informatics Lab, College of Computing and Data Science, Nanyang Technological University, Singapore, Singapore. Emails: ongz0070@e.ntu.edu.sg, ASJagath@ntu.edu.sg

thickness [3]. In addition, it also has promising applications in early diagnosis of Alzheimer's Disease (AD) where significant thinning of retinal nerve fiber layer and inner and outer macular rings were observed in AD subjects [4]. OCT is commonly used to image the retinal layers, which are layers of multiple structured layers of cells and synapses. These layers are shown in Fig. 1. However, manual ground truth of retinal layers is often challenging to obtain, and associated with potential inter and intra-observer variability. Moreover, speckle noise and blurring due to motion are typically present in OCT images [5], [6]. To reduce the need for manual labels, automatic OCT layer segmentation algorithms have been developed over the years.

Recently, deep learning methods have been proposed to be used for retinal layer segmentation and achieved state-of-the-art results. A number of these approaches are based on either convolutional neural networks [7], [8] or long short-term memory networks (LSTM) [9], which require large labeled datasets for training. However, label paucity due to the need for expert ophthalmologists input necessitates the need for semi-supervised approaches to reduce the annotation burden required. To reduce annotation need, multiple approaches, such as pseudo-labelling [10], vector quantization [11], and task and data level consistency supervision [12] were introduced. Clustering approaches [13] were also introduced in other areas such as intravascular OCT. In addition, as each retinal layer is extremely thin, sub-pixel accuracy is desired for retinal layer segmentation. Another feature of these retinal layers is that they have strict biological ordering. To address the need for sub-pixel level accuracy and anatomical constraints, He *et al.* [14] proposed a residual U-Net with two output branches to obtain pixel-wise layer segmentation probability prediction and a softmax mapping for probability distribution of boundary predictions. Topological ordering was imposed using an iterative topology module. SD-LayerNet [15], by Fazekas *et al.*, also ensures the correct topological ordering through the use of a fully differentiable topological engine. Adversarial learning has also been used for semi-supervised segmentation. SGNet consists of a segmentation and a discriminator network to utilize unlabeled data [16].

In this paper, we propose a novel generative adversarial learning for semi-supervised retinal layer segmentation, called GOctSeg. GOctSeg consists of a Generator, a Discriminator, and a Segmentor network, which is trained in an end-to-end manner. We hypothesize that by utilizing prior knowledge of

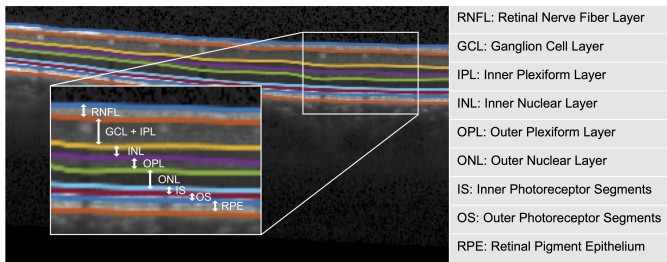

| | |
|---|---|
| RNFL: | Retinal Nerve Fiber Layer |
| GCL: | Ganglion Cell Layer |
| IPL: | Inner Plexiform Layer |
| INL: | Inner Nuclear Layer |
| OPL: | Outer Plexiform Layer |
| ONL: | Outer Nuclear Layer |
| IS: | Inner Photoreceptor Segments |
| OS: | Outer Photoreceptor Segments |
| RPE: | Retinal Pigment Epithelium |

Fig. 1. Retinal layers on an OCT B-scan. There are multiple retinal layers which can be captured with SD-OCT.

the biological ordering of retinal layers to generate synthetic labels, and the adversarial learning of the distribution of retinal images, GOctSeg requires less labeled data to attain reasonable performance for layer segmentation. The contributions of the paper are summarized as follows:

1) We propose a generative adversarial learning approach for semi-supervised retinal layer segmentation.
2) We propose to synthesize labels and incorporate the knowledge of the biological ordering of retinal layers for model training.
3) Instead of utilizing all the labels, we use a limited number of labels, namely, 6, 30, and 60 labels, for model training.
4) We investigated the performance of our approach on noisy and diseased images and demonstrated that our approach was able to perform consistently even in images with low signal-to-noise ratio or with blurred boundaries.

## II. METHODS

The proposed model (Fig. 2), named as GOctSeg, consists of a Generator $\mathcal{G}$, a Segmentor $\mathcal{S}$, and a Discriminator $\mathcal{D}$. $\mathcal{G}$ generates OCT images $I_s$ from an input synthetic layer label $Y_s$, where $s$ indicates that it is synthetic. $\mathcal{D}$ aims to differentiate the synthetic OCT images $I_s$ from real OCT images $I_r$, where $r$ indicates a real OCT image. $\mathcal{S}$ aims to segment the layers of $I_s$, which is expected to be the same as $Y_s$. Simultaneously, it aims to segment the layers of labeled OCT images $I_l$.

### A. Generator

$\mathcal{G}$ has a U-Net structure which consists of 4 blocks in the contracting path and 3 blocks in the expansive path. Each block consists of 2 convolutional layers with group normalization, ReLU, and dropout of 0.5 between them. The use of dropouts was added during experimentation to improve the model performance. The output layer is a $1 \times 1$ convolutional layer with 1 output channel.

### B. Discriminator

$\mathcal{D}$ tries to minimize the distribution differences between $I_s$ and $I_r$ through their encodings $\mathcal{D}(I_s)$ and $\mathcal{D}(I_r)$. This allows the synthetically generated image $I_s$ to be as similar as possible to an actual OCT image $I_r$ in terms of the appearance and texture. It consists of 5 convolutional layers, with 64, 128, 256,

512, and 1 output channels, respectively. The architecture is adapted from [17], with batch-normalization and LeakyReLU, except that there is no batch normalization between the first and second convolutional layer, and the last layer.

### C. Segmentor

$\mathcal{S}$ maps the synthetically generated image $I_s$ to predicted labels $\hat{Y}_s$ and $\hat{B}_s$, or image $I_l$ to $\hat{Y}_l$ and $\hat{B}_l$. In our implementation, each batch contains data from synthetic images $I_s$ and real-world labeled OCT images $I_l$ in a 1:1 ratio. The architecture of the Segmentor is given in Fig. 3. It comprises an Attention U-Net of 7 convolutional blocks, followed by two output branches. The first branch, conv-s, outputs a pixelwise segmentation of the different layers, whereas the second branch, conv-b, outputs $N$ boundary position distributions. Both branches, conv-s and conv-b, consist of a residual block followed by a $1 \times 1$ convolutional layer, similar to that in [14]. Each convolutional block consists of two $3 \times 3$ convolutional layers with group normalization, ReLU activation, and for the contracting path, a dropout of 0.5. In the expansive path, an *attention gate* module [18] is introduced.

*1) Gated Attention:* Let $l$ denote a convolution layer in the last 3 convolutional blocks ($l = 5, 6, 7$). Gated attention [18] utilizes the skip connection feature input $q^{8-l}$ for $5 \leq l \leq 7$ and the output from the previous block $o^{l-1}$ to generate an attention map $\alpha^l$. $o^{l-1}$ acts as a query to determine focus areas of $q^{8-l}$. The output $o^{l-1}$ consists of $N_{F,l-1}$ feature maps, where $F$ represents feature maps. A $1 \times 1$ convolution and group normalization are performed on $o^{l-1}$ to obtain $z^l$:

$$z^l = GroupNorm(W_o^l o^{l-1} + \beta_o^l) \qquad (1)$$

where $W_o^l \in R^{N_{F,l} \times N_{F,l-1} \times 1 \times 1}$ and $\beta_o^l$ denote the weight matrix and bias vector for the layer $l$. This is followed by an upsampling operation to obtain $\hat{z}^l$. Similarly, a $1 \times 1$ convolution and group normalization are performed on skip connection feature input, $q^{8-l}$, to obtain output $\hat{q}^{8-l}$:

$$\hat{q}^{8-l} = GroupNorm(W_q^{8-l} q^{8-l} + \beta_q^{8-l}) \qquad (2)$$

$W_q^{8-l} \in R^{N_{F,l} \times N_{F,8-l} \times 1 \times 1}$ and $\beta_q^{8-l}$ denote the skipped connection weights and biases, respectively.

The attention scores $\alpha^l$ are computed for convolutional block $l$ as

$$\alpha^l = \rho_1 \left( \text{BatchNorm} \left( W_\alpha^l (\max(0, (\hat{z}^l + \hat{q}^{8-l}))) + \beta_\alpha^l \right) \right).$$
$$(3)$$

where $W_\alpha^l \in R^{1 \times N_{F,l} \times 1 \times 1}$ are convolution weights with $1 \times 1$ kernel size. $\rho_1$ refers to the nonlinear sigmoid operation. The attention scores are then multiplied elementwise with the skip connection feature input, $q^{8-l}$, to identify salient features in the skip connection. The attention scores weigh the importance of the different parameters in $q^{8-l}$ to the final output.

$$q_\alpha^{8-l} = q^{8-l} * \alpha^l \qquad (4)$$

$*$ denotes element-wise multiplication. $o^{l-1}$ is upsampled from the previous layer using bilinear interpolation to obtain $\hat{o}^{l-1}$. Finally, $q_\alpha^{8-l}$ is concatenated with the upsampled output $\hat{o}^{l-1}$.

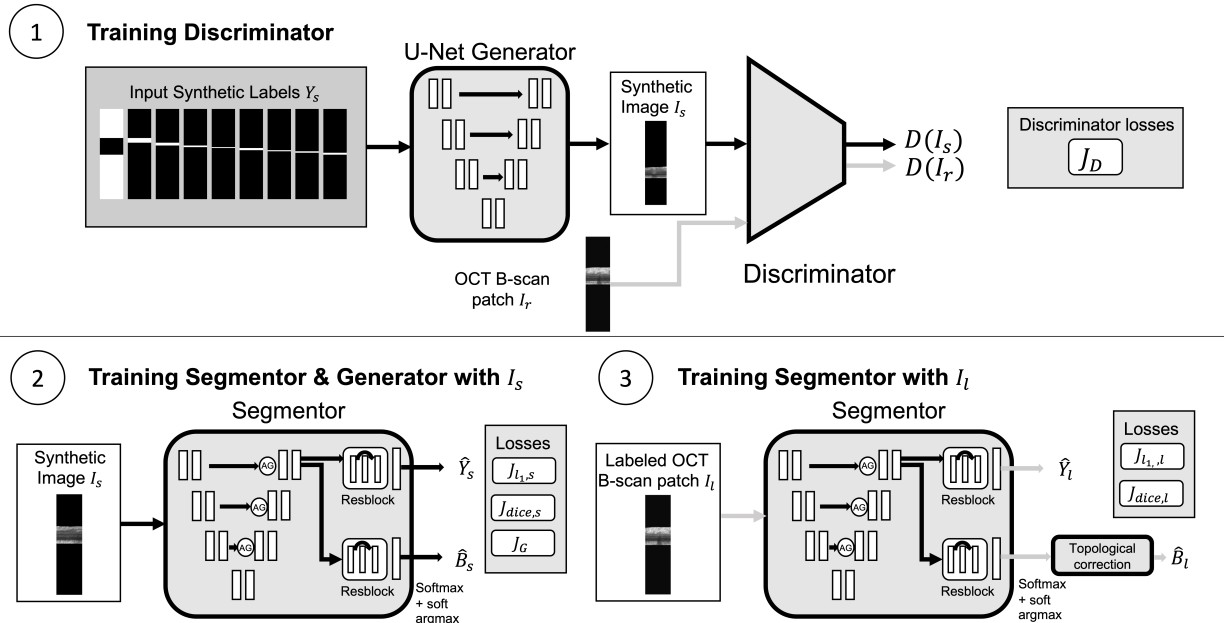

Fig. 2. Training procedure of GOctSeg. In each epoch, the Generator, Discriminator and Segmentor are updated. In the first stage, the Generator takes in synthetic label $Y_s$ as the input to generate $I_s$. The Discriminator takes in generated $I_s$ and $I_r$, an image from the training dataset, and computes the loss between the two distributions. Next, $I_s$ is used as input to the Segmentor, which is an Attention U-Net with two output branches to obtain $\hat{Y}_s$ and $\hat{B}_s$. As there is forward propagation through both the Generator and Segmentor during training, both networks are jointly trained by back-propagating both the segmentor and generator loss. Finally, $I_l$ from the labeled dataset is used as input to the Segmentor to obtain $\hat{Y}_l$ and $\hat{B}_l$. The model is updated with the supervised losses.

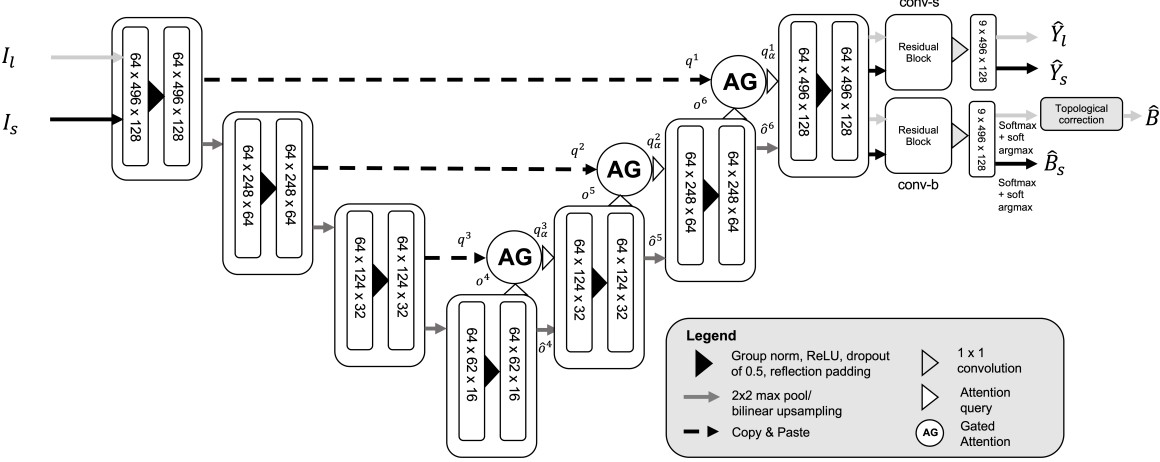

Fig. 3. Model architecture of Segmentor.

*2) Boundary Position Regression and Topological Correction:* In conv-b, the output from the Attention U-Net is passed into a residual block followed by a $1 \times 1$ convolutional layer with $N$ channels. A column-wise softmax is performed to obtain boundary position probabilities $p(\hat{B}; \theta)$ for input image $I$. The final boundary position $\hat{B}$ is obtained with soft argmax across J rows as in [14]: $\hat{B} = \sum_{j=1}^{J} j \cdot p(\hat{B}|I; \theta)$.

For real OCT images $I_l$, topological correction is imposed on $\{B^x\}_{x=1}^{x=X}$ for all $X$ A-scans to consider the anatomical constraint of the retinal layers [14], [15].

$$\hat{B}_{n+1}^x = \hat{B}_n^x + max(0, \hat{B}_{n+1}^x - \hat{B}_n^x), \quad (5)$$

where $n$ denotes the $n^{th}$ layer of the total $N$ layers.

## III. EXPERIMENTS

### A. Datasets and Data Preparation

Two independent datasets were used.

*1) DRMSHC:* The DRMSHC [14], [19], [20][1] consists of OCT volumetric scans of 14 healthy and 21 multiple sclerosis (MS) patients acquired using a Spectralis OCT system. Each macular cube scan consists of 49 B-scans, where 1 B-scan consists of 1024 A-scans and each A-scan has 496 pixels with a resolution of approximately 3.87 micrometers per pixel. Manual annotations delineate 8 layers, namely, Retinal Nerve Fiber Layer (RNFL), Ganglion Cell Layer (GCL) + Inner Plexiform Layer (IPL), Inner Nuclear Layer (INL), Outer Plexiform Layer (OPL), Outer Nuclear Layer (ONL), Inner Photoreceptor Segments (IS), Outer Photoreceptor Segments (OS), and Retinal Pigment Epithelium (RPE) resulting in 9 boundaries. The dataset was split into 12, 3, 20 patients for training, validation, and testing. The first 8 and 12 participants who are healthy and have multiple sclerosis respectively were selected for the test dataset, totaling 980 B-scans.

*2) Duke DME:* The Duke University DME dataset (Duke DME) [21] consists of 10 volumetric scans of patients with severe DME and acquired with the Spectralis device with an axial resolution of 3.9 micrometers per pixel and a lateral resolution of 11.07 to 11.59 micrometers per pixel. Each volumetric scan consists of 61 B-scans, of which 11 B-scans from 10 patients are annotated by 2 ophthalmologists and the rest are unlabeled. There are 7 layers delineated: RNFL, GCL + IPL, INL, OPL, ONL + IS, OS, and RPE. In addition, there are unannotated regions in certain A-scans for certain layers. 3-fold cross validation was used.

During training, each B-scan was cropped into images of $496 \times 128$ pixels with a stride of 64. For DRMSHC, this corresponded to 15 images, resulting in 8820 and 2205 images in the training and validation dataset, respectively. The whole DRMSHC training dataset was used as the $I_r$. For the Duke DME dataset, A-scans corresponding to regions where annotations were missing in the periphery of the labeled B-scans were removed. In total, there were 6 to 7 images per B-scan for the labeled dataset, corresponding to 748 images, and 11 images per B-scan for the unlabeled dataset, corresponding to 5500 images. The labeled B-scans were used as $I_l$ and both labeled and unlabeled B-scans were used as $I_r$.

Each B-scan, their ground truth label, and boundaries were flattened to the estimate of the Bruch's membrane (BM) and their background was removed [22]. The preprocessing steps were based on https://github.com/heyufan1995/oct_preprocess. They were shifted to a mean starting position of 248 pixels (half the height of the B-scan) and a standard deviation of 30 pixels, modelling on a normal distribution to add variability in vertical position. Training with full-width OCT B-scans was limited by the available RAM and cropping increased data size. At test time, the B-scan was not cropped to save time, but was flattened as a whole [23]. It was shifted vertically by 248 pixels, and the background was removed.

### B. Synthetic Labels Generation

Synthetic labels $Y_s$ and corresponding boundaries $B_s$ were generated during data preparation. As a simplification, the

---

[1]It is obtained from https://iacl.ece.jhu.edu/index.php/Resources

thickness of the *n*th retinal layer was modelled as a random normal variable $t_n$ with mean $\mu_{t_n}$ and standard deviation $\sigma_{t_n}$. During training, average thickness $\mu_{t_n}$ and the standard deviation $\sigma_{t_n}$ of the layer thickness were computed for the training dataset. Through computing the thickness of all layers and applying random vertical translations, we obtained $Y_s$. $B_s$ was generated by computing the vertical positions at each A-scan of the layer surfaces. The synthetic labels incorporated prior information regarding the positional ordering and thickness of the different retinal layers.

### C. Training Procedure

The model was trained in three steps. Firstly, the Discriminator was updated via the discriminator losses. Next, we computed the semi-supervised Segmentor losses $J_{l_1,s}$ and $J_{dice,s}$ and the Generator loss on the synthetic images. As the synthetic images from the Generator were used as inputs into the Segmentor, $J_{l1,s}$ and $J_{dice,s}$ were used to train the Generator as well via backpropagation. Finally, we finetuned the Segmentor with $J_{l1,l}$ and $J_{dice,l}$ on the labeled images.

*1) Generator Loss:* The generator loss minimizes the distribution differences between the generated synthetic images and real OCT images:

$$J_{\mathcal{G}} = \mathbb{E}_{Y \sim p_y(Y)}[(\mathcal{D}(\mathcal{G}(Y_s)) - r)^2] \qquad (6)$$

where $r$ represents whether the image is real, and is an array of 1.

*2) Segmentor Loss:* Assuming that $I_s$ falls within the distribution of $I_l$, the segmentation of $I_s$ to obtain predicted $\hat{Y}_s$ should be as similar as possible to the original synthetic labels $Y_s$. Similarly, the segmentation of $I_l$ to obtain predicted $\hat{Y}_l$ and should also be as similar as possible to the ground truth labels $Y_l$. We compute the segmentor losses as a combined Dice and Smooth $\ell_1$ loss.

The Dice loss can be computed between predicted $\hat{Y}_s$ and original synthetic labels $Y_s$, and between $\hat{Y}_l$ and $Y_l$, with $\epsilon$ as a smoothing constant. Let $v(j, x)$ be a pixel of $j$-th row and $x$-th A-scan, and $V$ be the set of all pixels. The Dice loss between $\hat{Y}_s$ and $Y_s$ is given as follows:

$$J_{dice,s} = \frac{1}{N_s + 1} \sum_{s=1}^{N_s+1} (1 - \frac{\epsilon + \sum_{v \in V} 2\hat{Y}_s(v)Y_s(v)}{\epsilon + \sum_{v \in V} Y_s(v) + \sum_{v \in V} \hat{Y}_s(v)}). \qquad (7)$$

The smooth $\ell_1$ loss can be used between $\hat{B}_s$ and the synthetic ground truth boundaries $B_s$, and between $\hat{B}_l$ and labeled ground truth boundaries $B_l$ [14]. The smooth $\ell_1$ loss between $\hat{B}_s$ and $B_s$ is given as follows:

$$J_{l1,s} =$$
$$\frac{1}{N_b X} \sum_{b=1}^{N_b} \sum_{x=1}^{N_x} (0.5(d^x)^2 \mathbb{1}(|d^x| < 1) + (|d^x| - 0.5)\mathbb{1}(|d^x| \geq 1)) \qquad (8)$$

where $d^x$ represents the difference between the predicted and the actual boundary positions for 1 A-scan.

The combined loss for the joint training of the generator and segmentor using synthetic images is computed by

$$J_s = \lambda_1 J_{dice,s} + \lambda_2 J_{\ell_1,s} + J_G. \qquad (9)$$

For the labeled images, it is computed by

$$J_l = \lambda_3 J_{dice,l} + J_{\ell_1,l} \qquad (10)$$

where $J_{dice}$ and $J_{\ell_1}$ denote the Dice loss and smooth $\ell_1$ loss respectively and $\lambda_1$, $\lambda_2$, $\lambda_3$ control the balance of the losses. $\lambda_1$, $\lambda_2$, $\lambda_3$ are weighing hyperparameters to be optimized.

*3) Discriminator Loss:* The loss function for the Discriminator is computed as

$$J_{\mathcal{D}} =$$
$$\frac{1}{2}\mathbb{E}_{I \sim p_{data}(I)}[(\mathcal{D}(I_r) - r)^2] + \frac{1}{2}\mathbb{E}_{Y \sim p_y(Y)}[(\mathcal{D}(\mathcal{G}(Y_s)) - s)^2] \qquad (11)$$

where $r$ and $s$ represent whether the image is real or synthetic via arrays of 1 or 0.

### D. Implementation Details

The models were trained on a NVIDIA Tesla V100 GPU, using Python 3.7.12 and PyTorch 1.11.0. Adam optimizer with a beta of 0.5 was used.

*1) DRMSHC:* Hyperparameter optimization was performed with $Tune$ library [24] using the training partition with 6 randomly selected labeled images as $I_l$. Random search of 15 trials with Async Successive Halving Algorithm (ASHA) scheduler was performed over the parameter space of learning rate $\lambda_{lr} = [1 \times 10^{-5}, 5 \times 10^{-4}, 1 \times 10^{-4}, 5 \times 10^{-3}, 1 \times 10^{-3}, 5 \times 10^{-2}, 1 \times 10^{-2}]$ and weighting parameters, $\lambda_1$ to $\lambda_3$ which was $[0.02, 0.1, 0.5, 1, 5, 10, 50]$ for 50 epochs. The hyperparameters and epoch that yielded the lowest mean absolute error (MAE) on the validation set were selected, which were: $\lambda_1 = 0.5$, $\lambda_2 = 0.5$, $\lambda_3 = 0.02$, $\lambda_{lr} = 0.001$, and epoch as 25. Next, we combined the validation and training set to train the model.

*2) Duke DME:* Nested 3-fold cross validation was performed, with the outer loop used to estimate the model generalization error and the inner loop used for hyperparameter optimization. In each inner loop, 15 random trials were performed over the same parameter space of $\lambda_{lr}$, $\lambda_1$, $\lambda_2$, and $\lambda_3$ as DRMSHC and the hyperparameters which yielded the lowest MAE on the validation fold were selected. As only 11 of 61 B-scans from each volumetric scan were labeled, both the labeled and unlabeled images were used to train the Discriminator. During training, A-scans with missing annotation were removed from the loss computation. After tuning, we combined the validation and training partition of inner loop to obtain the outer train set for each outer fold.

The images were augmented by random horizontal flipping with a probability of 0.5. Batch size of 8, with 4 labeled and 4 unlabeled images, was used. Instead of training the model using all labels, we selected the following quantities for training, with the rest of images used as $I_r$: (i) 6 labeled images; or (ii) 30 labeled images; or (iii) 60 labeled images, randomly selected from the whole labeled dataset.

## IV. RESULTS

At test time, only the Segmentor network was used to predict on the test dataset.

### A. Comparison with existing approaches

We implemented our approach on the DRMSHC dataset and compared with SD-LayerNet [15][2], a semi-supervised approach; the method by He *et al.* [14], which is fully supervised, and which we termed as Structured-Layer; and SGNet, another semi-supervised approach [16]. For comparison, we performed similar preprocessing steps involving retina flattening, background removal, and patch extraction for all methods. SD-LayerNet was implemented with 1 texture factor, batch size of 8 with 4 labeled and 4 unlabeled images, and gradient clipping of 0.5. To standardize for time and computational resources, we standardized the epochs for both SD-LayerNet and GOctSeg. For Structured-Layer, it was replicated based on [14], except with no vertical scaling, with gradient clipping of 0.5, same weight for pixels around the surfaces, and batch size of 4. Hyperparameter optimization determined the optimal epoch for convergence. For SGNet, the implementation was replicated based on [16], with lambda as 1, no random cropping, and gradient clipping of 20. The median frequency balance was computed based on the labeled data. Hyperparameter optimization determined the optimal learning rate and epoch to prevent instabilities. To obtain the MAE and root-mean-square error (RMSE) for SGNet, which is a segmentation approach, boundary tracing was done on the predictions. 3-fold cross validation was performed for all methods for Duke DME dataset. We evaluated all methods on the test dataset on MAE, RMSE, and Dice.

From Table I, it can be seen that our approach was statistically significantly better than other approaches for 6, 30 and 60 images using the one-tailed Wilcoxon signed-rank test for MAE and RMSE (p-values of comparison for 6 images for MAE: GOctSeg vs SD-LayerNet: $3.6 \times 10^{-162}$, GOctSeg vs SGNet: $4.8 \times 10^{-162}$, GOctSeg vs Structured-Layer: $3.0 \times 10^{-162}$). Using Dice, we observed that Structured-Layer performed better for 30 and 60 images, as semantic segmentation is an easier task than boundary regression. When the experiment for Structured-Layer was repeated with the full labeled dataset, we obtained $3.78 \mu m$ for MAE, which is almost equivalent to the MAE of GOctSeg for 60 labeled images.

When we computed the MAE by the different boundaries as seen in Fig. 4, our approach performed comparably across all boundaries, other than the boundary between RNFL and GCL+IPL. In contrast, Structured Layer trained on 6 labeled images had very large mean MAE for many boundaries. Although SD-LayerNet also performed reasonably well, there was a higher mean MAE for all boundaries with 6 labeled images, for all boundaries except upper boundary of RNFL for 30 images, and for all boundaries with 60 labeled images. While GOctSeg should perform better with 30 labeled images,

---

[2]The implementation of SD-LayerNet used was https://github.com/ABotond/SD-LayerNet.

TABLE I
COMPARISON OF GOCTSEG WITH SD-LAYERNET, SGNET, AND
STRUCTURED-LAYER ON THE DRMSHC DATASET FOR 6, 30, AND 60
LABELED IMAGES. AXIAL RESOLUTION WAS 3.87 MICROMETERS PER
PIXEL (MPP). THE * INDICATES A P-VALUE ≤ 0.05 WHEN COMPARING
WITH GOCTSEG USING THE ONE-SIDED WILCOXON SIGNED-RANK TEST.

| Metric | Method | 6 | 30 | 60 |
|---|---|---|---|---|
| MAE | GO | 4.25 ± 3.08 | 4.94 ± 7.73 | 3.77 ± 2.83 |
| | SDLN | 7.38 ± 6.18* | 6.32 ± 5.50* | 4.85 ± 3.22* |
| | SGN | 7.55 ± 8.00* | 7.39 ± 6.86* | 7.64 ± 7.63* |
| | SL | 35.82 ± 16.75* | 8.73 ± 5.04* | 7.18 ± 4.72* |
| RMSE | GO | 6.22 ± 5.54 | 6.91 ± 9.12 | 5.42 ± 5.00 |
| | SDLN | 13.93 ± 12.47* | 11.27 ± 11.25* | 7.97 ± 7.23* |
| | SGN | 12.62 ± 19.42* | 11.52 ± 15.33* | 11.94 ± 15.86* |
| | SL | 39.64 ± 16.97* | 10.39 ± 6.49* | 8.25 ± 5.08* |
| Dice [a] | GO | 0.86 ± 0.07 | 0.85 ± 0.12 | 0.87 ± 0.07 |
| | SDLN | 0.82 ± 0.10* | 0.84 ± 0.09* | 0.86 ± 0.08* |
| | SGN | 0.84 ± 0.09* | 0.84 ± 0.09* | 0.83 ± 0.09* |
| | SL | 0.44 ± 0.35* | 0.86 ± 0.07 | 0.89 ± 0.06 |
| Dice [b] | GO | 0.87 ± 0.07 | 0.86 ± 0.12 | 0.88 ± 0.07 |
| | SDLN | 0.82 ± 0.10* | 0.84 ± 0.09* | 0.86 ± 0.08* |
| | SGN | 0.79 ± 0.10* | 0.79 ± 0.10* | 0.79 ± 0.11* |
| | SL | 0.21 ± 0.21* | 0.60 ± 0.36* | 0.64 ± 0.37* |

[a] Dice computed from predicted semantic segmentations.
[b] Dice computed from predicted boundaries.
GO: GOctSeg, SDLN: SD-LayerNet, SGN: SGNet, SL: Structured-Layer.

the slightly higher MAE was due to outliers in the predicted segmentation for a minority of the images. When comparing the 50th percentile of mean MAE of B-scan, we obtained 3.85$\mu$m with 6 images, compared to 3.74$\mu$m with 30 images.

*B. Ablation studies*

We also conducted two ablation studies, (i) training the Segmentor network in a fully supervised manner (GOctSeg-Sup) and (ii) training the Generator and Disciminator first to generate synthetic images, followed by training the Segmentor on the synthetic and labeled images (GOctSeg-Sep). Table II shows the model performance of GOctSeg, GOctSeg-Sup, and GOctSeg-Sep. We observed that our approach had the lowest mean MAE and RMSE in the 6 and 60 images setting (p-values of MAE for 6 images' comparison: GOctSeg vs GOctSeg-Sup: $3.0 \times 10^{-162}$, GOctSeg vs GOctSeg-Sep: $2.3 \times 10^{-137}$). Using one-sided Wilcoxon signed-rank test, we obtained statistical significance of $p = 1.1 \times 10^{-6}$ when comparing GOctSeg-Sep and GOctSeg for 30 labeled images, with GOctSeg having more predictions with a lower MAE. Upon investigation, the 50th percentile of mean MAE per b-scan for GOctSeg-Sep with 30 labeled images was 3.81$\mu$m compared to 3.74$\mu$m for GOctSeg.

*C. Implementation on Duke DME dataset*

The Duke DME dataset contains eyes with severe DME pathology. Table III shows a comparison of the mean metric and range of GOctSeg, SD-LayerNet, SGNet, and Structured-Layer on the Duke DME dataset for 6, 30, and 60 labeled images for 3-fold cross validation. Our approach performed significantly better than SD-LayerNet, SGNet, and Structured Layer for 6 labeled images on MAE and RMSE, as seen from Table III. From Fig. 5 we observed that GOctSeg had smoother segmentation compared to SD-LayerNet at 6 images. For the

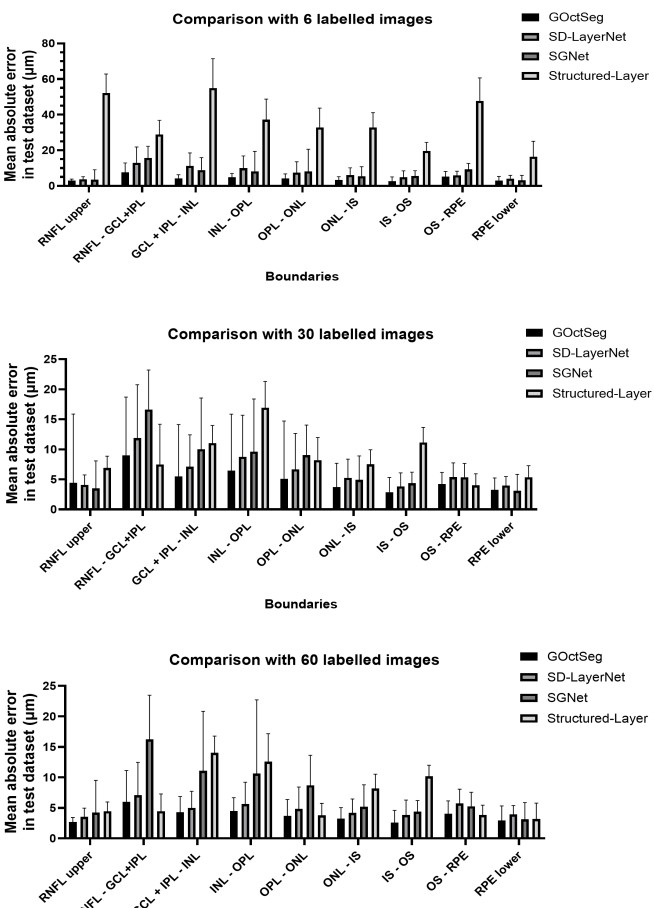

Fig. 4. Bar plot of the MAE ($\mu$m) by boundary for DRMSHC dataset with 6, 30, and 60 labeled images. Error bar represents the standard deviation. x-axis represents the 9 boundaries, where RNFL upper, RNFL-GCL+IPL, and RPE lower indicate the upper boundary of RNFL, boundary between RNFL and GCL+IPL, and lower boundary of RPE respectively.

limited number of labeled images, Structured Layer appeared to be unable to delineate the layers properly, but was able to do so with enough labeled images, i.e. 60 images.

*D. Comparison under noisy conditions*

Next, we investigated the performance of GOctSeg with SD-LayerNet under different noisy scenarios at 6 images. We compared with SD-LayerNet as it had the closest performance to GOctSeg. In the first three scenarios, we investigated reducing the signal-to-noise ratio of the images by approximately 10%, 20%, and 40% (10% red, 20% red, and 40% red on Table IV). This was done by adding Gaussian noise with mean of 0 and standard deviation, empirically determined by calculating the decrease in signal-to-noise ratio via adding varying amounts of Gaussian noise, to the volumetric scans and then clipping the images from 0 to 1. In the next two scenarios, we investigated adding Gaussian noise only at the high frequency components for boundary blurring. In the fourth scenario (i.e. FT on Table IV), we performed Fourier transform on the image and adding Gaussian noise at the periphery of the real component of the

TABLE II
ABLATION STUDIES ON THE DRMSHC DATASET FOR 6, 30, AND 60 LABELED IMAGES. * INDICATES A P-VALUE ≤ 0.05 WHEN COMPARING WITH GOCTSEG USING THE ONE-SIDED WILCOXON SIGNED-RANK TEST.

| Metric | Method | 6 | 30 | 60 |
|---|---|---|---|---|
| MAE | GO | 4.25 ± 3.08 | 4.94 ± 7.73 | 3.77 ± 2.83 |
| | GO-Sup | 14.07 ± 10.45* | 7.61 ± 5.56* | 9.28 ± 6.69* |
| | GO-Sep | 5.48 ± 4.83* | 4.42 ± 3.83* | 6.32 ± 5.02* |
| RMSE | GO | 6.22 ± 5.54 | 6.91 ± 9.12 | 5.42 ± 5.00 |
| | GO-Sup | 16.10 ± 11.28* | 9.56 ± 6.66* | 10.71 ± 7.00* |
| | GO-Sep | 7.82 ± 7.44* | 6.44 ± 6.20* | 8.64 ± 7.32* |
| Dice [a] | GO | 0.86 ± 0.07 | 0.85 ± 0.12 | 0.87 ± 0.07 |
| | GO-Sup | 0.50 ± 0.17* | 0.81 ± 0.12* | 0.85 ± 0.09* |
| | GO-Sep | 0.83 ± 0.09* | 0.86 ± 0.08 | 0.79 ± 0.12* |
| Dice [b] | GO | 0.87 ± 0.07 | 0.86 ± 0.12 | 0.88 ± 0.07 |
| | GO-Sup | 0.50 ± 0.33* | 0.69 ± 0.29* | 0.61 ± 0.37* |
| | GO-Sep | 0.84 ± 0.10* | 0.87 ± 0.08* | 0.81 ± 0.12* |

[a] Dice computed from predicted semantic segmentations.
[b] Dice computed from predicted boundaries.
GO: GOctSeg, GO-Sup: GOctSeg-Sup, GO-Sep: GOctSeg-Sep.

TABLE III
COMPARISON OF GOCTSEG WITH SD-LAYERNET, SGNET, AND STRUCTURED-LAYER ON DUKE DME DATASET FOR 6, 30, AND 60 LABELED IMAGES WITH 3-FOLD CROSS VALIDATION. AXIAL RESOLUTION WAS 3.9 MPP.

| Metric | Method | 6 | 30 | 60 |
|---|---|---|---|---|
| MAE | GO | 13.7 [12.8,15.4] | 11.3 [9.0,15.0] | 8.0 [5.3,10.0] |
| | SDLN | 17.6 [12.8,23.7] | 9.2 [7.0,11.6] | 8.4 [6.2,10.2] |
| | SGN | 19.3 [13.4,30.4] | 11.3 [9.8,12.1] | 10.7 [9.8,11.5] |
| | SL | 31.1 [26.7,36.2] | 13.6 [10.2,16.6] | 11.1 [8.2,16.6] |
| RMSE | GO | 21.1 [20.5,22.4] | 17.3 [15.5,20.3] | 12.6 [7.8,15.4] |
| | SDLN | 27.1 [20.5,36.1] | 14.9 [11.7,18.1] | 13.6 [10.2,15.7] |
| | SGN | 28.9 [19.2,43.8] | 16.7 [13.9,19.0] | 15.8 [13.9,17.3] |
| | SL | 35.7 [31.8,38.8] | 17.2 [12.9,20.3] | 14.8 [11.1,20.8] |
| Dice [a] | GO | 0.73 [0.72,0.74] | 0.76 [0.65,0.82] | 0.83 [0.79,0.87] |
| | SDLN | 0.73 [0.72,0.75] | 0.82 [0.78,0.85] | 0.83 [0.80,0.86] |
| | SGN | 0.72 [0.65,0.78] | 0.78 [0.77,0.79] | 0.79 [0.79,0.80] |
| | SL | 0.54 [0.44,0.62] | 0.82 [0.81,0.84] | 0.84 [0.82,0.86] |
| Dice [b] | GO | 0.74 [0.73,0.76] | 0.76 [0.67,0.80] | 0.83 [0.80,0.87] |
| | SDLN | 0.71 [0.67,0.75] | 0.81 [0.77,0.85] | 0.82 [0.79,0.86] |
| | SGN | 0.72 [0.67,0.77] | 0.77 [0.76,0.79] | 0.78 [0.77,0.79] |
| | SL | 0.27 [0.18,0.33] | 0.57 [0.45,0.69] | 0.72 [0.53,0.82] |

[a] Dice computed from predicted semantic segmentations.
[b] Dice computed from predicted boundaries.
GO: GOctSeg, SDLN: SD-LayerNet, SGN: SGNet, SL: Structured-Layer.

complex image. We performed inverse Fourier transform to get the new images. In the fifth scenario (i.e. HPF on Table IV), we performed high pass filtering of the image, added Gaussian noise of mean 0 and sigma 0.05 to the high pass filtered image, which was added with the low pass filtered image.

As shown on Table IV which compares the models over various noisy conditions, GOctSeg performed robustly in most noisy conditions and performed better in 20% SNR reduction, 10% SNR reduction, and boundary blurring conditions in MAE, RMSE and Dice compared to SD-LayerNet.

## V. DISCUSSION AND CONCLUSION

In this paper, we introduced a semi-supervised learning generative adversarial learning framework for retinal layer segmentation. Via our methodology, we propose the generation of synthetic images from synthetic boundaries, which would be able to capture positional and intensity changes of the

TABLE IV
COMPARISON OF GOCTSEG WITH SD-LAYERNET ON DUKE DME DATASET FOR 6 LABELED IMAGES UNDER DIFFERENT NOISY CONDITIONS WITH 3-FOLD CROSS VALIDATION.

| Metric | Method | GO | SDLN |
|---|---|---|---|
| MAE | 40% red | 17.3 [12.8,23.3] | 16.1 [13.0,17.9] |
| | 20% red | 14.7 [11.4,19.3] | 16.2 [13.4,18.3] |
| | 10% red | 12.0 [10.5,12.9] | 17.6 [12.5,20.5] |
| | FT | 12.2 [10.3,15.0] | 16.7 [11.2,21.3] |
| | HPF | 12.1 [10.2,14.6] | 15.0 [11.6,16.8] |
| RMSE | 40% red | 26.2 [20.2,37.0] | 25.7 [20.6,29.4] |
| | 20% red | 23.9 [18.3,30.1] | 25.2 [20.8,29.1] |
| | 10% red | 19.2 [17.4,20.2] | 27.7 [19.4,34.1] |
| | FT | 18.4 [15.9,21.3] | 25.8 [17.6,31.5] |
| | HPF | 18.8 [16.1,22.2] | 23.3 [18.2,26.6] |
| Dice [a] | 40% red | 0.72 [0.69,0.75] | 0.72 [0.70,0.74] |
| | 20% red | 0.75 [0.72,0.76] | 0.73 [0.72,0.74] |
| | 10% red | 0.76 [0.74,0.78] | 0.71 [0.66,0.74] |
| | FT | 0.74 [0.72,0.75] | 0.72 [0.66,0.77] |
| | HPF | 0.75 [0.72,0.78] | 0.73 [0.70,0.76] |
| Dice [b] | 40% red | 0.70 [0.65,0.75] | 0.72 [0.69,0.75] |
| | 20% red | 0.76 [0.73,0.78] | 0.71 [0.69,0.74] |
| | 10% red | 0.78 [0.76,0.80] | 0.70 [0.65,0.74] |
| | FT | 0.76 [0.73,0.78] | 0.71 [0.66,0.77] |
| | HPF | 0.76 [0.73,0.79] | 0.72 [0.68,0.77] |

[a] Dice computed from predicted semantic segmentations.
[b] Dice computed from predicted boundaries.
GO: GOctSeg, SDLN: SD-LayerNet.

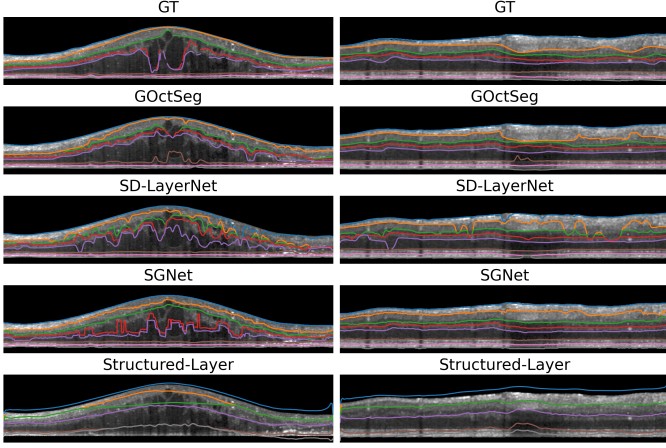

Fig. 5. Visualization of predicted boundaries on 2 B-scans from different volumetric scans from Duke DME test dataset. Background is cropped for visualization and images resized. From top to bottom: Ground Truth (GT), predicted boundaries from GOctSeg, SD-LayerNet, SGNet, and Structured-Layer at 6 labeled images.

different layers. This allowed us to constraint the search space for mapping the image to the corresponding label, reducing the number of labels required for a good segmentation. As demonstrated above, when labels were limited, GOctSeg was able to perform almost on par to Structured-Layer trained on the full labeled dataset for DRMSHC. In addition, we demonstrated that GOctSeg was able to perform better than other approaches with 6 labeled images, in severe disease pathology cases. In such cases, the presence of pathologies such as intraretinal fluid would complicate the convergence of the algorithm due to unclear intensity changes in the retinal

layers. Nonetheless, this did not significantly degrade the performance of GOctSeg. Furthermore, GOctSeg was robust in different noisy conditions. In particular, we compared the performance of GOctSeg with SD-LayerNet after reducing the signal-to-noise ratio and blurring boundary, and observed that it performed better compared to SD-LayerNet in most conditions. This is extremely promising in real-world scenarios, where OCT images usually have low signal-to-noise ratio [25].

Nevertheless, there are still limitations. Currently, preprocessing is required, to minimize the variability of the retina images for modeling the distribution with the Generator. The ground truth annotations provided might also have imperfections, although such slight imperfections do not affect the synthetic labels as the labels were generated based on the mean thickness and the standard deviations of the ground truth. One future improvement would be to generate more sophisticated labels which would capture curvature changes of the retina.

In summary, GOctSeg would be extremely useful in obtaining a reasonable segmentation when labels are limited. Furthermore, given an estimate of the layer thickness, it is also possible to utilize the model to explore images in the absence of ground truth as the Segmentor could be trained on synthetically generated images. This could occur in situations where only the caliper measurements of the retina are available, which provides an estimate of the layer thickness but lacks the granularity offered by layer segmentation. In such scenarios, the model could be trained to generate synthetic images which could be used to learn the intensity and biological ordering of these layers. This would be critical in real world settings.

## ACKNOWLEDGMENT

We thank Dr. Cheng Jun (A*STAR Singapore) for assistance and comments on the manuscript and useful discussions.

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
