# OpenReview forum: "Generative adversarial learning for semi-supervised retinal layer segmentation in OCT images"
_IEEE.org/EMBS/BHI/2024/Conference — IEEE BHI'24_

### Official Review · Reviewer_LaGr · 2024-08-10
**My Review for IEEE EMBS BHI 2024 Submission 100**

**Overall Rating:** 5
**Confidence:** 4

**Other Quality Metrics:**

(a) Clarity of writing: fair

(b) Clinical Significance: good

(c) Methodological Novelty: fair

(d) Experiments and Results: fair/good

**Questions For The Authors:**

See above.

**Strengths:**

The developed network is bringing together existing approaches.

**Summary Of The Paper:**

The paper uses GAN for the segmentation of the retinal layer in OCT images.

**Weaknesses:**

Why not use n-fold cross validation?
It seems difficult to understand that the MAE and RMSE results are significantly different (based on the very large standard deviations). Including the actual p-values might help the reader. Some more details on how Wilcoxon was used would be of interest.

---

### Official Review · Reviewer_4xEH · 2024-08-11
**An effective semi-supervised learning approach for retina boundary segmentation combining both metrics the DICE factors and smoothness for evaluation**

**Overall Rating:** 7
**Confidence:** 4

**Other Quality Metrics:**

(a) Clarity of writing - good
(b) Clinical Significance - great
(c) Methodological Novelty - good
(d) Experiments and Results - great

**Questions For The Authors:**

STOTA
- Many more studies are related to semi-supervised boundary segmentation, e.g., for IVOCT image data. It might be reasonable to extend your references.

Methods:
- You mention that your learning approach is trained in an end-to-end fashion. Please emphasize more clearly: How are the different parts (Generator, Discriminator, and Segmentor) connected during training? In Fig. 2 it seems like the networks are trained separately.
- How did you derive the "prior information" for the synthetic labels? Please state a reference or describe your manual approach.
- Did you evaluate the performances with a cross-validation?
- Please explain the difference between the chosen hyperparameters for the DME and DRMHC datasets. In particular, \lambda_1 and \lambda_3 are chosen much higher for DME. Considering Eq. (9) and (10) the DICE coefficients are weighted higher - is there a specific reason to do so? Are the layers thicker/thinner in the DME dataset?

Results and Discussion:
- Use boundary names or abbreviations introduced in Fig. 1 for errorbar plots in Fig. 4?
- Use the same y-axis scaling in Fig. 4? Or, at least, increase the number of ticks for the scenario with 6 labelled images.
- Ablation study: What data set sizes are used for the different approaches (-Sup, -Sep) in comparison to your best approach?
- Improve the layout of results Tables.

**Strengths:**

- Complex methodology presented to improve the boundary segmentation
- Extensive experimental evaluation on two different data sets

**Summary Of The Paper:**

The authors present an improved semi-supervised learning approach for retinal boundary segmentation. The architecture consists of three main parts, a Generator, Segmentor, and Discriminator trained end-to-end. More specifically, the authors combine metrics describing the smoothness and DICE of the segmentation to adjust the training. In the experimental studies, the effectiveness of the approach is presented with significant improvements compared to other architectures. Here, the authors differentiate between the number of labelled images used for training and demonstrate the increasing performance for all the different boundary layers.

**Weaknesses:**

- A cross-validation study could strengthen the outcome of the presented results
- The discussion is a little short compared to the in part very detailed method sections

---

### Official Review · Reviewer_MQcZ · 2024-08-14
**A paper with solid results but may lack of novelty and generalizability**

**Overall Rating:** 6
**Confidence:** 4

**Other Quality Metrics:**

(a) Clarity of writing; fair
(b) Clinical Significance; good
(c) Methodological Novelty; fair
(d) Experiments and Results good

**Questions For The Authors:**

See above weakness.

**Strengths:**

Weakly supervised study is of practical importance.

The results section seems solid.

**Summary Of The Paper:**

This paper proposed a semi-supervised learning framework for retinal layer segmentation. Relying upon the order of retinal layers, the authors used the conditional generative model to generate the synthetic retinal images based on the corresponding synthetic labels. Then, a segmentor is trained on the synthetic dataset and fine-tuned through the real-world data. The model is evaluated on two OCT datasets.

**Weaknesses:**

-- This study relies on a strong assumption that the retinal tissues have a clear, ordered layer structure that can be segmented based on intensity differences. The reviewer is unclear about its generalization capability on tissues with non-layered structures.

-- The performance of the study ties heavily to the quality of the generated synthetic images. As such, is this study still validated on diseased models?

-- The idea of using the conditional generative model to get synthetic data to enrich the training set (data augmentation) is not very novel.

-- B_s and B_y seem undefined.

-- Some algorithm implementation details can be moved from the Methods section into the Experiments Section to improve the presentation of the paper.

---

### Decision · Program_Chairs · 2024-09-23

Accept